# SOLAR ECLIPSE-INDUCED PERTURBATIONS AT MID-LATITUDE DURING THE 21 AUGUST 2017 EVENT

Bolarinwa J. Adekoya[1], Babatunde O. Adebesin[2], Timothy W. David[1], Stephen O. Ikubanni[2], Shola J. Adebiyi[2], Olawale. S. Bolaji[3,4] and Victor. U. Chukwuma[1]

[1]Department of Physics, Olabisi Onabanjo University, P.M.B. 2002, Ago Iwoye, Nigeria

[2]*Space Weather Group, Department of Physical Sciences, Landmark University, P.M.B 1001, Omu-Aran,*

*Kwara State, Nigeria*

[3]*Department of Physics, University of Lagos, Akoka – Yaba, Lagos, Nigeria*

[4]*Department of Physics, University of Tasmania, Hobart, Australia*

*Correspondence to*: Bolarinwa J. Adekoya (adekoyabolrinwa@yahoo.com; adekoya.bolarinwa@oouagoiwoye.edu.ng)

## Abstract

A study of the response of some ionospheric parameters and their relationship in describing the behaviour of ionospheric mechanisms during the solar eclipse of 21 August 2017 is presented. Mid-latitude stations located along the eclipse path and with data availability on the Global Ionospheric radio Observatory (GIRO) database were selected. The percentage of obscuration at these stations range between 63-100%. Decrease in electron density during the eclipse is attributed to reduction in solar radiation and natural gas heating. The maximum magnitude of the eclipse coincided with $hmF2$ increase and with a lagged maximum decrease in $NmF2$ consistently at the stations investigated. The results revealed that the horizontal neutral wind flow is as a consequence of the changes in the thermospheric and diffusion processes. The unusual increase/decrease in the shape/thickness parameters during the eclipse period relative to the control days points to the perturbation caused by the solar eclipse. The relationship of the bottomside ionosphere and the F2 layer parameters with respect to the scale height are shown in the present work as viable parameters for probing the topside ionosphere during eclipse. Furthermore, this study shows that in addition to traditional ways of analysing the thermospheric composition and neutral wind flow, proper relation of standardized $NmF2$ and $hmF2$ can be conveniently used to describe the mechanisms.

**Keywords:** solar eclipse; solar radiation; bottomside profile parameters; $NmF2$ and $hmF2$; Topside ionosphere; GIRO database.

## 1  Introduction

Solar eclipse provides opportunity to study the causes of drastic changes in the atmosphere arising from reduction in solar radiation and plasma flux. The atmosphere responded to these changes by modifying the electrodynamic processes and ionization supply of its species to the nighttime-like characteristics during the daytime. Different physical mechanisms (e.g. neutral wind, thermospheric composition, diffusion process etc.) that explain the distribution of plasma at the different ionospheric layers are well established. However, these mechanisms do compete with themselves in explaining the ionosphere, especially the topside ionosphere (see Gulyaeva, 2011).

At mid-latitudes, the effect of diffusion processes and its relationship with the thermospheric compositions has been extensively studied during episodes of solar eclipse (Muller-Wodarg et al., 1998; Jakowski et al., 2008; Le et al., 2009; Wang et al., 2010; Chuo, 2013). At equatorial and low-latitude regions, the $E \times B$ plasma drift had been used to explain the circumstances of solar eclipse on transport processes (Adeniyi et al., 2007; Adekoya et al., 2015). Recently, attention has been drawn to the study of the topside ionosphere during an eclipse for improved prediction and modelling (Huba and Drob, 2017; Chrniak and Zakharenkova, 2018). Reinisch et al., (2018) compared the modelled and measured studies of electron densities at the altitude range of about 150 - 400 km during the eclipse. They found that at lower altitude (at about 150 km) the modelled and the measured agreed well to the changes in the altitude profile of electron density compared to at higher altitudes. The authors however posited that it would be improved if the model *NmF2* peak falls more slowly to better match the data. Consequently, the present study investigates the effects of solar eclipse of August 21, 2017 on the constituents of the ionosphere at mid-latitudes using some ionosonde data (bottomside parameters, scale height (H) estimated from the fitted α-Chapman layer) which have not been given much attention in previous works especially in analysing solar eclipse effect. Using these parameters to analyse the circumstances of solar eclipse at the topside ionosphere and its plasma distribution mechanisms make this paper significantly different from previous studies. Thus, we intend to achieve by analysing the ionospheric parameters that controls the distribution of plasma at the topside and bottomside layers of the F2 region. To shed light on these analysis, section 2 highlights the data source, methodology, and path of the eclipse. The results and discussion were presented in section 3, while section 4 presents the summary and concluding remark of the result.

## 2   The solar eclipse path and Data source

With regards to the eclipse of 21 August 2017, the totality of the eclipse is visible from within a narrow corridor that traverses the United States of America. However, in the surrounding areas, which include all of mainland United States and Canada, the eclipse was partial. From the footprint of the Moon's shadow as seen from some locations, the eclipse started from around 17:00 UT and ended around 20:00 UT. Figure 1 shows the detail coverage area and circumstances of the solar eclipse. More details of its path can be seen via NASA – Total solar eclipse of 2017 August 21 (https://eclipse.gsfc.nasa.gov/). The details on the local circumstances of the eclipse, the time of the first, mid and last contact of the eclipse over the ionosphere of the investigated stations was highlighted in table 1. More details on the total solar eclipse event and its partiality, the circumstances surrounding its progression and its magnitude of obscuration can be obtained through the link http://xjubier.free.fr/en/index_en.html. The path of the eclipse informed the choice of stations. The ionospheric data used for this study for the selected mid-latitude stations were obtained from the Global Ionospheric Radio Observatory (GIRO) networks, http://giro.uml.edu/ (Reinisch and Galkin 2011) and manually validate. The calculated daily average of summation Kp, Ap and solar flux indices was


## 3  Methods of data analysis

*NmF2* values for both the eclipse and control days were obtained from their corresponding critical
frequencies (*foF2*) using the expression: *NmF2* = ((*foF2*)$^2$ / 80.5) e/m$^3$. The control day value is the average
value of the two days before/after the eclipse day (i.e. 6, 12, 24 and 27). These reference days were chosen
such that they have similar geomagnetic, interplanetary and solar properties with the eclipse day. The daily
average value of control days and eclipse day interplanetary index (Ap and Kp), and solar flux unit index
(F10.7) ranges from 8 – 12 nT for Ap, 2 – 3 for Kp index and 75.6 – 89.1 sfu (1 solar flux unit (sfu) = 10$^{-22}$
Wm$^{-2}$ Hz$^{-1}$) for F10.7, indicating that geomagnetic and solar activities of these days is unsettled (see
Adekoya et al., 2015 for classification of geomagnetic activity). The typical behaviour of the *NmF2* and
*hmF2* on the eclipse day (i.e. *NmF2*e and *hmF2*e) was compared with that of the control day (*NmF2*c and
*hmF2*c) to observe the changes brought by the short period of loss of photoionization in the ionosphere.
This will measure the direct consequence of the solar radiation disruption (due to the eclipse) on the
ionospheric chemical, transport and thermal processes in the F2 layer. The ionized layer depends majorly
on three parameters, viz: *NmF2, hmF2*, and the plasma scale height ($H_m$).

The GIRO provides access autoscaled values of ionospheric parameters generated by Automatic Real-Time
Ionogram Scaler with True height (ARTIST) algorithm, which is inherent in the UMLCAR-SAO Explorer
(Reinisch and Huang 1983; Galkin et al., 2008; Reinisch and Galkin 2011), facilitates the derivation of
bottomside profiles. From the ULMCAR-SAO Explorer, the manually scaled ionogram with high accuracy are
calculated from the standard true-height inversion program (Reinisch and Huang, 1983; Huang and
Reinisch, 1996). The parameters obtained include the critical frequency (*foF2, Hz*), and its height (*hmF2,*
*km*) of the F layer and the shape (*B1*), and thickness (*B0*) parameters. Likewise, the scale height ($H_m$) of the
F2 layer is obtained from the bottomside. It is estimated from the fitted α-Chapman function with a
variable scale height, *H*(*h*), to the measured bottomside profile *N*(*h*), which then determined as the
Chapman scale height at *hmF2* (i.e. *H*(h > *hmF2*) ≈ $H_m$ (*hmF2*)) (Huang and Renisch 2001; Reinisch and
Huang 2001; Reinisch et al., 2004). The topside profile is then related to the scale height at the layer, from
the bottomside profile, represented with α-Chapman function (Reinisch and Huang, 2001). This is because
the Chapman function described the electron density profile, N(h) aptly. Also, $H_m$ provides a linkage
between the bottomside ionosphere and the topside profiles of the F region (Liu et al., 2007). Therefore,
$H_m$ describes the constituents of the ionospheric plasma, which decreases with increasing altitude. The
fitting formulas of α-Chapman function are provided in equation 1 below.
$$N_e = N_m F2 exp\left\{\frac{1}{2}[1 - z - exp(-z)]\right\}; \qquad z = \frac{h - hmF2}{H_m} \qquad \qquad 1$$

where all the parameters have their usual meaning.

However, Xu et al. (2013) and Gulyaeva (2011) related ionospheric F2 - layer scale height, $H$ to the topside
base scale height, $Hsc$, given by $Hsc = hsc\text{-}hmF2 \approx 3 \times H_m$). Where $hsc$ is the height at which the electron
density of the F2-layer falls by a factor of an exponent, at an upper limit of 400 km altitude (i.e. $NmF2$/e)
(see Xu et al., 2013). That is, the region where electron density profile gradient is relatively low. Gulyaeva
(2011) showed theoretically that $Hsc$ increase over Hm by a factor of approximately three (3) and is a
consequence of the $Ne$/$NmF2$ ratio ($Ne$ – plasma density), which corresponds to $H_m$ in the Chapman layer.
At altitudes very close to $hmF2$, the ratio equals 0.832, while it is 0.368 at altitudes beyond the $hmF2$.
Therefore, we adopted the definition of Gulyaeva (2011) for the topside base scale height as the region of
the ionosphere between the F2-peak and 400 km altitude. Summarily, the topside based scale height
ionosphere here is defined as the region between the F2 peak and $hsc$ or $3H_m$. It is thus evident that H is a
key and essential parameter in the continuity equation for deriving the production rate at different
altitudes, a pointer to the F2 topside electron profiler, as well as a good parameter for evaluating the
transport term (Yonezawa, 1966; Huang and Reinisch, 2001; Reinisch and Huang, 2001; Belehaki et al.,
2006; Reinisch et al., 2004). Consequently, the parameter $H_m$ can be used as a proxy for observation
relating to the topmost side electron density profile. Furthermore, the division of the topsides and the
bottomside ionosphere may be related to the difference in the effective physical mechanisms in the
regions. Hence, the bottomside parameters $B1$ and $B0$ of the ionosphere, as presented in this work, helped
in examining the perturbation of solar eclipse in the bottomside ionospheric F2 layer.

**4 Results and Discussion**
This section presents the temporal evolution of the maximum electron density ($NmF2$), and its
corresponding height ($hmF2$) over the ionosphere at the selected mid-latitude stations along the path of
solar eclipse of 21 August 2017. The control day variation relative to the eclipse day is also presented.
Figure 2 presents the variation of maximum electron density and the corresponding peak height, during
both the eclipse and control days. Figure 3 depicts the variation of scale height and the bottomside
parameters ($B0$ and $B1$) due to the eclipse by superposing plots for both the eclipse and control days.
Analysis of these parameters during an eclipse event may help in the modelling of the ionospheric profiles
(the topsides and bottomside electron density distribution profile) during the short nighttime-like period of
the day.

Figure 2a presents the $NmF2$ and $hmF2$ variations during the eclipse event and the control day over the
Idaho National Lab; having an obscuration magnitude of 100% around the daytime period. The effect of the
disruption of solar radiation was evident as the $NmF2$ started decreasing at the first contact of the eclipse
compared to an incessant increase on the control day in Fig. 2ai. The start time or first contact (08:43:31
LT), the maximum magnitude period (10:01:53 LT) and the end time or the last contact (11:25:46 LT) of the
eclipse are marked with the vertical lines S, M and E respectively. The decrement in *NmF2* during the
eclipse phase was due to reduction in the ionization. This reduction caused changes in the photochemical
and transport process of the atmosphere during the daytime, thus exhibiting nighttime characteristics. It
should be noted that the maximum decrease in *NmF2* did not coincide with the maximum magnitude of the
eclipse obscuration, rather with a time lag of few minutes, i.e., 1030 LT. This lag period fell within the
relaxation period over Idaho ionosphere, with *NmF2* and *hmF2* simultaneously attaining their peak
magnitudes of 1.67 $e/m^3$ and ~ 239 km. Hence, the ionosphere returned to its pre-eclipse state. Contrary to
the decrease in the *NmF2* amplitude at the recovery phase of the eclipse, the *hmF2* increases, attained 239
km peak around 1030 LT and then decreases depicting the eclipse caused morphology.

The ionosphere over Boulder, Eglin AFB, Austin, Millstone Hill and Point Arguello did not show any contrary
variation to that observed over Idaho during the eclipse event. The decrease and increase in *NmF2* and
*hmF2* after the maximum magnitude are simultaneous. The only exception was that the local time at which
each station observed the effects were different. Their obscuration percentage ranged from 62.5 – 93.37%.
This did not cause any significant change in the way they responded to the reduction in solar heating. The
ionosphere over Boulder experienced the totality of the eclipse with 93.37 % magnitude, which is next to
Idaho (100%) in obscuration, the *hmF2* was observed to increase few minutes after the maximum
magnitude of the obscuration. This behaviour is typical for other stations at the eclipse window, but the
time of *NmF2* minimum decrease did not always coincides with the *hmF2* enhancement after the maximum
obscuration. These observations posit that the minimum rate of electron production does not necessarily
translate to the peak electron density of the molecular gases formed. This is because the electron
concentration depends on the loss rate by dissociative recombination, too.

At mid-latitudes, the ionospheric F2 plasma distribution is controlled by diffusion processes (Rishbeth
1968). There are two basic mechanisms that define the diffusion process during an eclipse: First is the
coolness brought by the partial removal of photoionization (Müller-Wodarg et al., 1998), which is believed
to instigates the downward diffusion process, and the atmospheric expansion due to the gradual increase
in the temperature after the totality. The downward diffusion process was related to the increase in the
molecular gas ($N_2$) concentration during the cooling process. However, the aftermath of the coolness was
related to the upward diffusion process. These mechanisms were proxy to the electron density distribution
during the eclipse window. Our analysis suggests that the observed decrease in *NmF2* is due to the
downward diffusion flux of the plasma while the increase that followed is by upward diffusion (e.g. Le et al.,
2009; Adekoya and Chukwuma 2016). Several works on eclipse (Müller-Wodarg et al., 1998; Grigorenko et
al., 2008; Adekoya and Chukwuma 2016; Hoque et al., 2016) have shown that it was not just the electron
density that is being affected during an eclipse window, but the thermospheric wind as well, since the
thermospheric wind emanating from the ratio of gas species is related to the variation in electron density.
It has been observed that the increase in the mean molecular gas of thermospheric composition decreases
the electron density and vice versa. Le et al. (2010) related the valley of electron density distribution during
the eclipse phases to the contraction/compression and expansion of the atmosphere brought by the
decrease and increase in temperature. Chukwuma and Adekoya (2016) attributed the decrease in the
electron temperature to the downward vertical transport process and the decrease in the cooling process
to the upward vertical transport process.

Figure 3 describes the variation of $H_m$, $B1$ and $B0$ in three columns respectively for all the stations. Looking
at the $H_m$ plots, one can see that there was a define morphological description of $H_m$ at the eclipse window.
From the first contact of the eclipse, there was an incessant increase in peak variation that maximized some
minutes after the maximum contact of the eclipse, i.e., about 15 – 45 mins later. Following the peak
magnitude after the maximum contact of the eclipse, the $H_m$ sharply decreases, reaching the minimum
peak before its rather increase throughout the remaining period of the eclipse second phase. It was further
observed that the minimum decrease in $NmF2$ amplitude corresponds to increase in $H_m$ at all stations;
implying the upward lifting of the topside electron to the region of higher altitude at the eclipse window.
Hence, the scale height variation highlights the decrease in electron production and the vertical distance
through which the pressure gradient falls at the topside during the eclipse activity. The observation
illustrates the mutual relationship between the $NmF2$ and $H_m$, which may aid in extrapolating the topside
ionospheric profile (Gulyaeva, 2011). In essence, scale height changes observed during the eclipse window
can be used to explain the pressure gradient, electron density distribution and transport processes. In this
sense, the diffusion coefficients are expressed as ratio of determinants (determinant here refers to the
concentration of species ([O] and [N$_2$]), with the size of the determinants depending upon both the number
of species in the gas mixture and the level of approximation. Therefore, the increase (decrease) in the scale
height can be used as a proxy for downward (upward) diffusion process at the topside ionosphere.
Consequently, the thermospheric wind, which causes plasma distribution in the topside ionosphere, is
induced by solar radiation. Moreover, the significant changes observed in the scale height variation during
the eclipse window also indicated that transport processes are affected as they are temperature
dependent. Therefore, changes in the thermospheric compositions due to the solar eclipse at the topside
layer will affect the density profiles of the ionosphere.

It is noteworthy that the increase (decrease) in the scale height decreases (increases) the electron density
during the eclipse window. The sensitivity of electron density to temperature at the topside directly affects
the electron density profile (e.g. Wang et al., 2010); as cooling due to decrease in temperature results in
decrease in the electron density via reduced ionization. This indicates that the decrease (increase) in
electron temperature at the topside ionosphere causes the increase (decrease) in the scale height, which is
related to the diffusion and transport processes and subsequently affect the pressure gradient of the
plasma. From plots of $H_m$ (fig. 3) and $NmF2$ (fig. 2), it was observed that the minimum decrease in $NmF2$
corresponded with peak increase in scale height. This implies that the topside ionosphere is more sensitive
(than the bottomside) to any changes in the solar radiation. Thus, the pressure gradients can be analysed in
terms of either the scale height or electron density during solar eclipse.

From column 2 and 3 of Figure 3, we observed that the measured shape ($B1$) and thickness ($B0$) parameters
of the ionosphere over these stations exhibit significant variations during the eclipse event. $B1$ responded
with a decrease at the first contact of the eclipse compared to the control day. This decrease was gradual
throughout the eclipse window and followed the variation of solar ionizing radiation. However, $B0$ variation
differs to that of the $B1$ observation. The $B0$ increases from the first contact and reached the maximum
peak few minutes after the maximum obscuration magnitude, which coincided with the minimum decrease
in $B0$. Generally, the pattern of the day to day variation of the bottomside parameters was the average
morphology, but the increase in the $B0$ and the decrease in the $B1$ parameters during the eclipse period
compared to the control day was a notable one and can be related to the perturbation caused by the solar
eclipse. During the eclipse, the solar radiation was lost; trapped atomic ions $O^+$ was converted into
molecular ion ($NO^+$ and $O_2^+$) by charge transfer, owing to the sufficient concentration of molecular gasses
($N_2$ and $O_2$) (Rishbeth, 1988). The height of the ionospheric slab indeed increased with reduced width,
which is attributable to compression due to loss of solar heating.

The behaviour of the ionosphere can be explained during solar eclipse with any of the components that
constitute the topside and the bottomside ionosphere and can be looked at, from the angle of the
percentage of concentration of the components. In this regard, the deviation percentage of $NmF2$ ($\delta NmF2$)
and $hmF2$ ($\delta hmF2$) during the eclipse day away from the control day were plotted in Figure 4. This is done
to describe the contribution of the thermospheric wind and compositions. Although observing the variation
of $NmF2$ and $hmF2$ alone can be used for observing the changes in the behaviour of the thermospheric
compositions and wind flow, if properly analysed, but it is more convenient to describe these mechanisms
by standardizing the original variables used during the event. The normalization effort (with the use of
$\delta NmF2$ and $\delta hmF2$) presents the original variation of $NmF2$ and $hmF2$ onto directions which maximize the
variance. Consequently, the result can be used for analyses of any mechanisms that drive the ionospheric
plasma, if properly related.

The deviation percentage in Figure 4 was defined as the ratio of (($NmF2e − NmF2c$)/$NmF2c$) x 100. The
same relation is defined for the $hmF2$ parameter. As earlier pointed out, during eclipse period, neutral
composition becomes the dominant chemical process arising from diffusion activities. The increase in the
neutral composition leads to the increase in the molecular gas concentration and compete with diffusion
process. Hence the deviation percentage discusses the neutral composition changes and delineate how
these changes may affect the electron densities as well as its profiles in the atmosphere during the eclipse.
The respective maximum and minimum peak response of the deviation percentage is attributed to the
enhancement and depletion of $δNmF2$. One can sees from the plots, the deviation percentage started
increasing at the first contact of the eclipse (the first dashed vertical line) and reached the maximum,
appearing few minutes after the maximum magnitude of the eclipse (the second dashed vertical line). This
behaviour is similar to the conditions of the neutral compositions during the eclipse event reported by
Muller-Wodarg et al. (1998).

Another important process observed in this study is the neutral wind flow effect. To identify the direction
of the wind, the $δNmF2$ colour legend in the contour plots was used in Figure 4. The negative values
represent a westward wind contribution and the positive values is for the eastward wind. Looking at the
marked eclipse region in the figure, it was revealed that the $δNmF2$ started decreasing from the first
contact of the eclipse, maximized few minutes after the maximum contact mark and, thereafter decreases.
It has been established that at daytime, the peak height of the plasma will be reduced due to lost in
recombination. At nighttime, equatorward neutral wind drives the F2-layer plasma to higher altitudes
where recombination rate is slower. The ionospheric processes during solar eclipse is said to represent a
partial nighttime/sunset ionospheric process (Adekoya et al., 2015; Adekoya and Chukwuma, 2016). Thus,
the F2 plasma behaviour at the eclipse window is induced by the equatorward neutral wind flow. The
neutral wind acts jointly with the plasma flows from the topside ionosphere, resulting in F2 region plasma
density variation. Therefore, the westward/eastward neutral wind flow is related to the
depletion/enhancement in the deviation, which was clearly shown in the marked eclipse region of the
figure. The plots in Figure 3 had established the ionospheric dynamics of diffusion processes, neutral
compositions and the flow of neutral wind caused by the eclipse perturbation, which can invariably reduce
the effectiveness and reliability of radio wave propagation.

Relative to the mutual relationship between the topside and bottomside ionosphere, we considered the
linear correlation coefficient ($R$) of $H_m$ versus $hmF2$ and H versus $B0$ during the eclipse window, In Fig. 5., $R$
ranges from (0.80 - 0.90) for $H_m$/$hmF2$ relationship, and 0.57-0.89 for the $H_m$/$B0$ connection. This good
linear agreement revealed the dependence of $hmF2$ and $B0$ on the scale height. Apart from revealing the
dependence between the parameters, the relationship may also provide a convenient way for modelling

the topside profile from the knowledge of the bottomside parameter, *B0*, during the eclipse period. Further, fig. 6 Illustrates the relationship between the bottomside (continuous line) and the topside (dashed line) ionosphere over Idaho National Lab during solar eclipse compared to the non-eclipse period. On the left side was the ionospheric profile during the first contact of the eclipse, the middle and right-side profiles are during the maximum contact and last contact of the eclipse respectively. The black curve represents the profile for the eclipse day (August 21) and the red curve is for the one of the selected reference days, August 27. It is clear from the plots that the ionospheric profiles vary with the solar ionizing radiation at the eclipse window and shows the suitability of using the bottomside F-region for probing the topside ionosphere. This behaviour was typical for the ionospheric profiles from other stations along the path of the eclipse. Also, the strong correlation between *hmF2* and $H_m$ indicates that there may be some interrelated physical mechanisms controlling the behaviour of the plasma at the topside ionosphere during solar eclipse. That is, *hmF2* is strongly depends on neutral wind flow and explain the state of thermospheric compositions (e. g. Liu et al., 2006; Fisher et al., 2015). Since all these parameters competes during the eclipse, one can argue that with the accessibility of one, in place of the other (as a consequence of their relationship), the prediction and modelling of the ionosphere can be conveniently achieved.

**5 Conclusions**

This paper presents the induced perturbation of solar eclipse of 21 August 2017 on the ionospheric F parameters and how they describe the mechanisms of the ionosphere at mid-latitude. The perturbation effects and dynamics during a solar eclipse episode using ionospheric F2 parameters (*NmF2* and *hmF2*), the bottomside profile thickness (*B0*) and shape (*B1*) parameters of electron density and the plasma scale height ($H_m$), which are not often used for eclipse study, were investigated. These parameters represent the state of the F-region ionosphere. The changes observed during the eclipse phase is related to the reduction in solar radiation and natural gas heating. The *NmF2* minimum was attained around 30 - 45 minutes after the totality of the eclipse when it decreases to about 65% of its control day. This decrease in *NmF2* was uplifted to the higher altitude where recombinational rate is reduce compared to the non-eclipse day. The thickness and shape parameters which are often limited to the bottomside F-region were seen as viable parameters for probing the topside ionosphere, relative to the scale height during the eclipse. Therefore, their relationship in describing one another is established. The implication is that eclipse-caused perturbation could have been better explained using some ionosonde parameters. The changes in the neutral wind flow, thermospheric compositions and diffusion processes found their explanation in the behaviour of the F region plasma during eclipse. In addition, it can be concluded that the behaviour of δ*NmF2* and δhmF2 during eclipse can be conveniently used to describe the mechanisms of thermospheric composition and wind flow.

**Acknowledgements**

We acknowledge use of global ionospheric Radio Observatory data provided by ULMCAR (http://ulcar.uml.edu/DIDBase/) and the World Data Center for Geomagnetism, Kyoto (http://wdc.kugi.kyoto-u.ac.jp/index.html) for geomagnetic activity data. We thank the management team of the national Aeronatics and Space Administration (NASA) service (http://eclipse.gsfc.nasa.gov) and http://xjubier.free.fr/en/site_pages/SolarEclipseCalc_Diagram.html for progression and eclipse local circumstances information. The authors thank Professor Ljiljana R, Cander and the anonymous reviewers for their constructive corrections and suggestions that tremendously improved the structure and quality of the paper.

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

**Table Caption**

**Table 1:** List of ionosonde station, geographic coordinate, eclipse progression time and percentage of maximum obscuration.

**Figure Captions**

**Figure 1:** The orthographic map showing the coverage area and circumstances of the solar eclipse, and the observatory stations of the total solar eclipse event of August 21, 2017 . The thick blue line region of represents the path of the maximum magnitude of the eclipse and the pale blue lines mark the region of where the partial eclipse is experienced, with the magnitude of partiality.

**Figure 2:** Ionospheric *NmF2* and *hmF2* variations during the eclipse day (black continuous line) and the control day (dash blue line). The three vertical lines represents the different phases of the eclipse (S - start time of the initial phase, M - the period of the maximum magnitude of the eclipse, and E - the end time of the recovery phase or the last contact of the eclipse progression). The local time of the respective eclipse contact points for each station are given in table 1.

**Figure 3:** The local time variation of the ionospheric scale height and the bottomside (*B0* and *B1*). The other features are the same as in Fig. 1.

**Figure 4:** Variation of the deviation percentage of *NmF2* (δ*NmF2*) and *hmF2* (δ*hmF2*) magnitudes for observing the changes in the behaviour of the thermospheric composition and wind flow related to the loss rate during the eclipse phase. The three vertical dashed lines marked the eclipse start time, the time of maximum obscuration and the last contact time of the eclipse (i.e. eclipse phase). Table 1 highlights the local time contact point of the eclipse corresponding the international standard time (IST) eclipse progression. The direction of wind was identify using the δNmF2 colour legend, the negative values represents the westward wind direction and the positive values is for the eastward wind.

**Figure 5:** Linear relationship of H versus *hmF2* and H versus *B0* during the eclipse of 21 August 2017 progression phase.

**Figure 6:** Example of the ionospheric profile at the eclipse window of Idaho National Lab showing the bottomside profile (continuous line) and the modelled topside profile shown as a dashed line. The maximum point of the continuous line is the point in which the peak value of the measured foF2 and hmF2 are obtained. The respective measured values foF2, hmF2 and the corresponding B1, B0, and Hm parameters values are provided in the plot. The black curve represents the profile for the eclipse day (August 21) and the red curve is for the one of the selected reference days, August 27. On the left side, was the profile during the first contact of the eclipse, the middle and the right profiles are for the maximum contact and the last contact of the eclipse respectively.

**Table 1:** List of ionosonde station, geographic coordinate, eclipse progression time (Universal time/ Local
time) and percentage of maximum obscuration.

| Station | GLat | GLong | Eclipse Start time (UT)/(LT) | Eclipse Max Time (UT)/(LT) | Eclipse End Time (UT)/(LT) | % of max obscuration | UT to LT difference |
|---------|------|-------|------------------------------|----------------------------|----------------------------|----------------------|---------------------|
| IDAHO NATIONAL LAB | 43.81 | 247.32 | 16:14:15/ 08:43:31 | 17:32:37/ 10:01:53 | 18:56:30/ 11:25:46 | 100 | 16:29:17 |
| BOULDER | 40 | 254.7 | 16:22:33/ 09:21:21 | 17:46:10/ 10:44:58 | 19:13:46/ 12:12:34 | 93.37 | 16:58:48 |
| EGLIN AFB | 30.5 | 273.5 | 17:04:41/ 11:18:29 | 18:37:08/ 12:50:56 | 20:03:48/ 14:17:36 | 83.322 | 18:13:48 |
| AUSTIN | 30.4 | 262.3 | 16:40:45/ 10:09:55 | 18:10:10/ 11:39:20 | 19:39:35/ 13:08:45 | 65.93 | 17:29:10 |
| POINT ARGUELLO | 34.8 | 239.5 | 16:02:39/ 08:00:15 | 17:16:55/ 09:14:31 | 18:39:36/ 10:37:12 | 64.608 | 15:57:36 |
| MILLSTONE HILL | 42.6 | 288.5 | 17:27:28/ 12:41:16 | 18:45:53/ 13:59:41 | 19:58:38/ 15:12:26 | 62.533 | 19:13:48 |













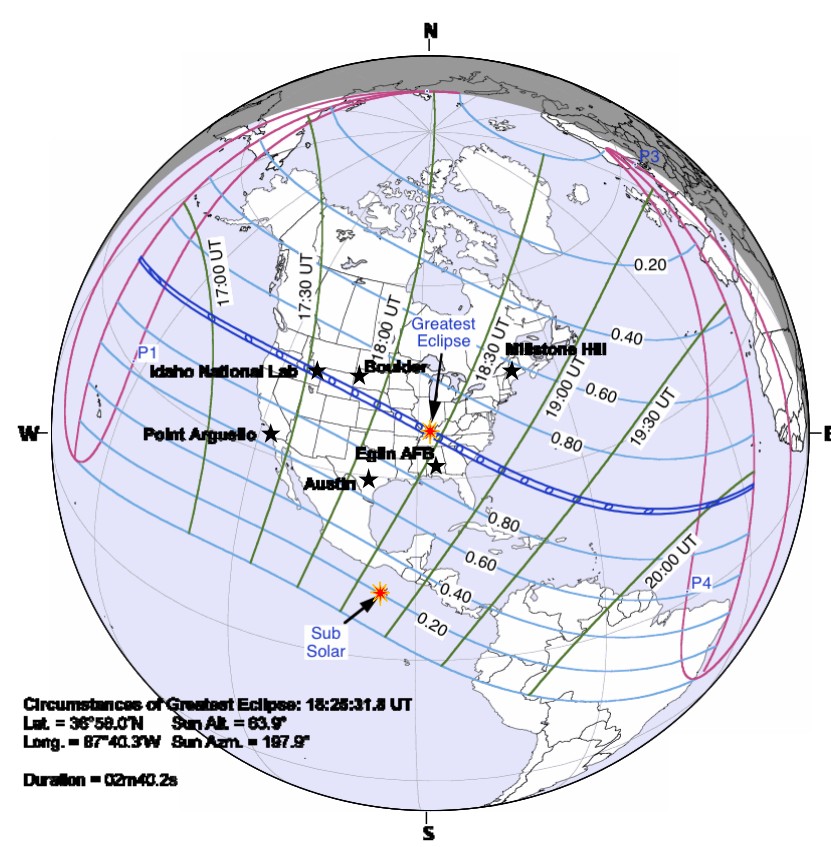


**Figure 1:** The orthographic map showing the coverage area and circumstances of the solar eclipse, and the observatory stations of the total solar eclipse event of August 21, 2017 . The thick blue line region of represents the path of the maximum magnitude of the eclipse and the pale blue lines mark the region of where the partial eclipse is experienced, with the magnitude of partiality.

557

558

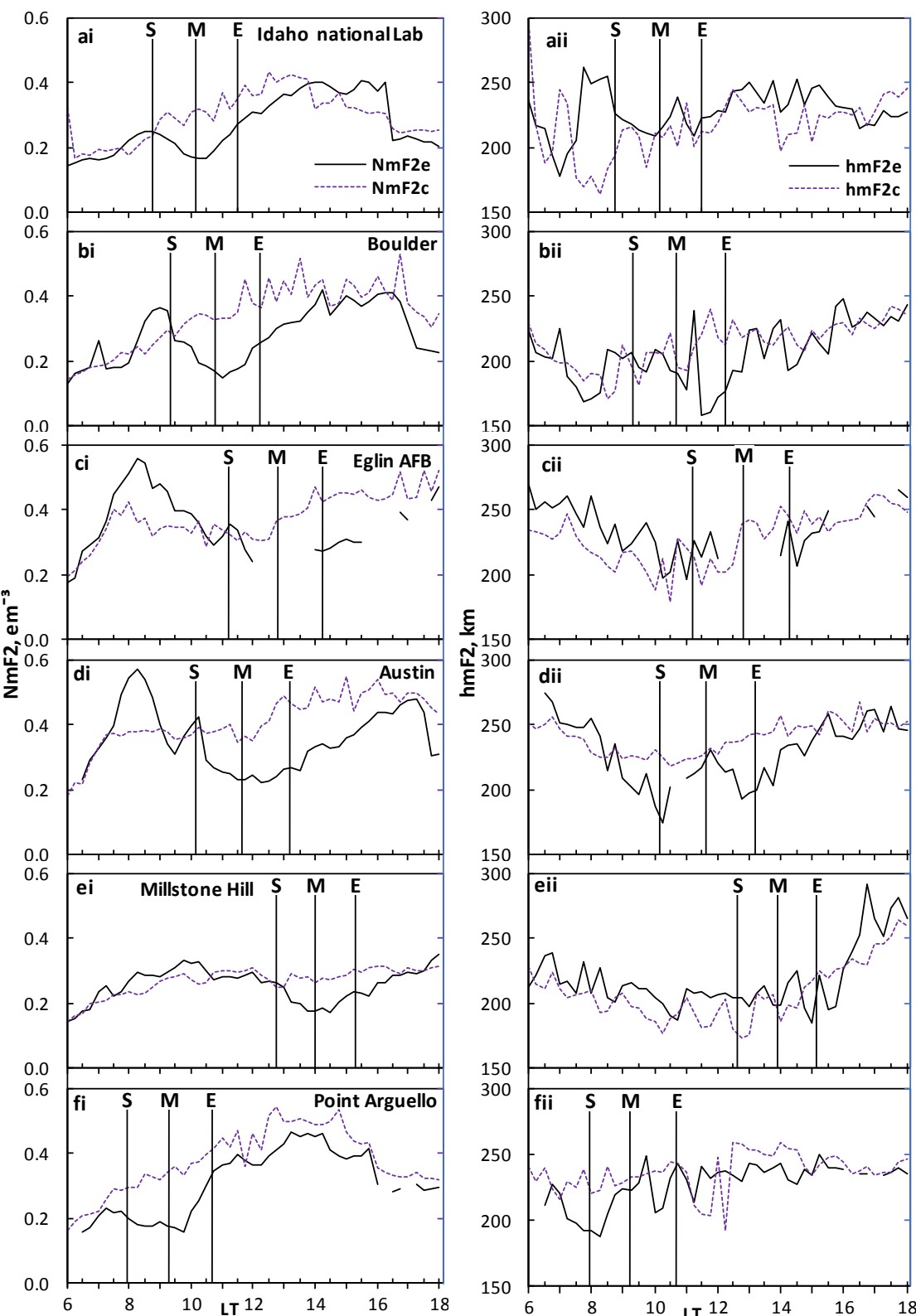

559

**Figure 2:** Ionospheric *NmF2* and *hmF2* variations during the eclipse day (black continuous line) and the control day (dash blue line) was presented to delineate effect of solar eclipse of August 21, 2017 on the ionosphere. The three vertical lines represents the different phases of the eclipse (S - start time of the initial phase, M - the period of the maximum magnitude of the eclipse, and E - the end time of the recovery phase or the last contact of the eclipse progression). The local time of the respective eclipse contact points for each station are given in table 1.

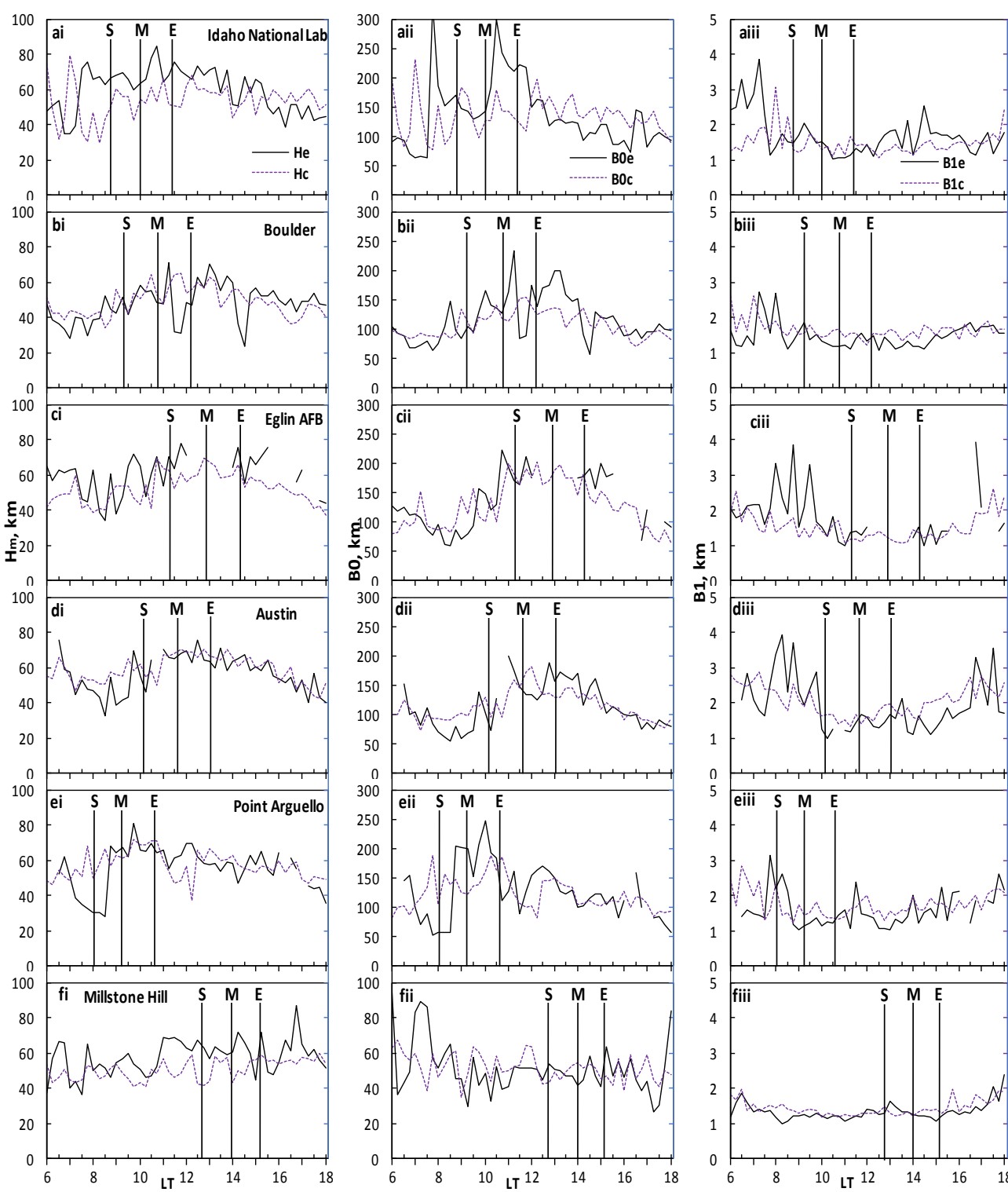


**Figure 3:** The local time variation of the ionospheric scale height and the bottomside (*B0* and *B1*). The other features are the same as in Fig. 1.


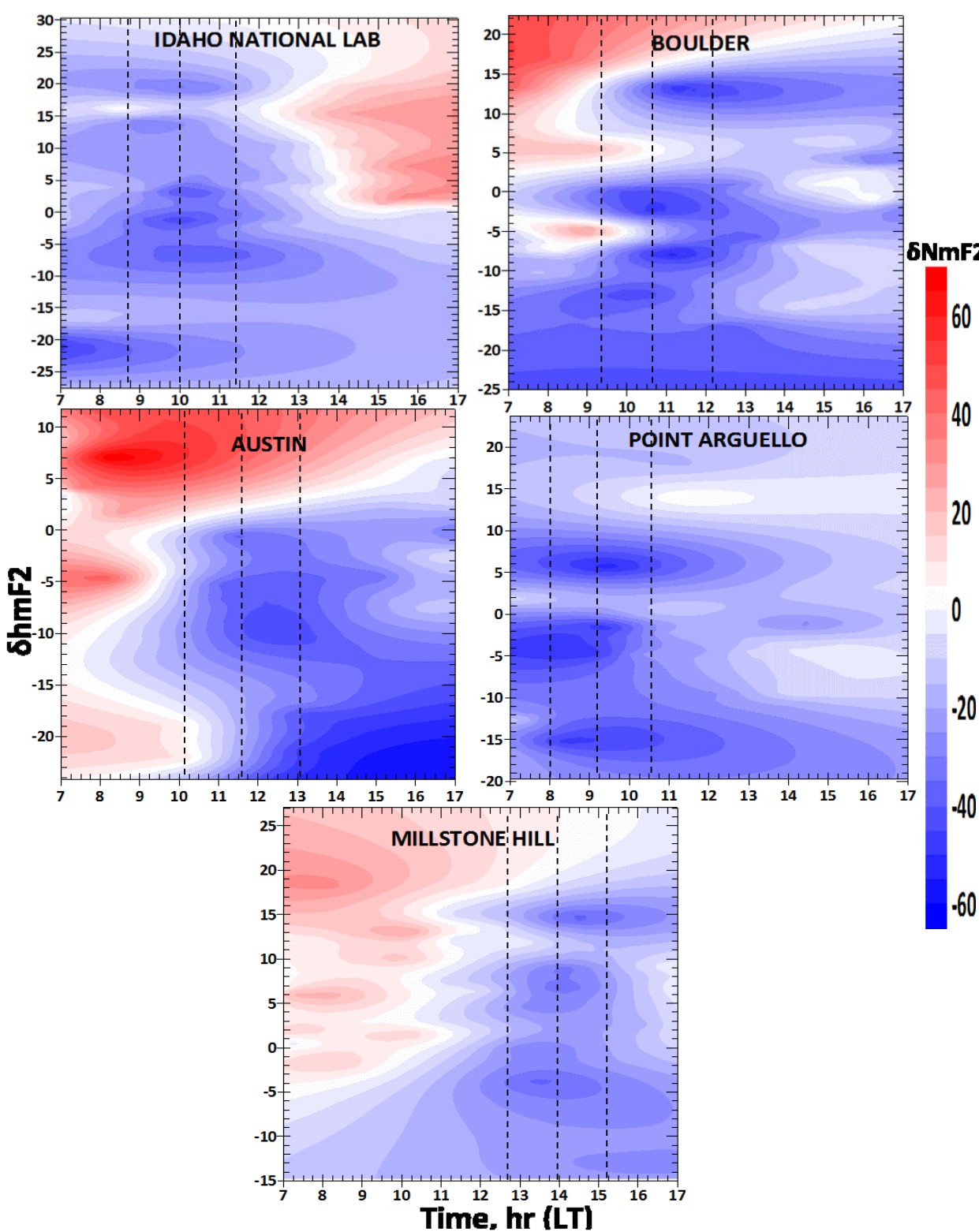


**Figure 4:** Variation of the deviation percentage of *NmF2* (*δNmF2*) and *hmF2* (*δhmF2*) magnitudes for observing the changes in the behaviour of the thermospheric composition and wind flow related to the loss rate during the eclipse phase. The three vertical dashed lines marked the eclipse start time, the time of maximum obscuration and the last contact time of the eclipse (i.e. eclipse phase). Table 1 highlights the local time contact point of the eclipse corresponding the international standard time (IST) eclipse progression. The direction of wind was identify using the δNmF2 colour legend, the negative values represents the westward wind direction and the positive values is for the eastward wind.

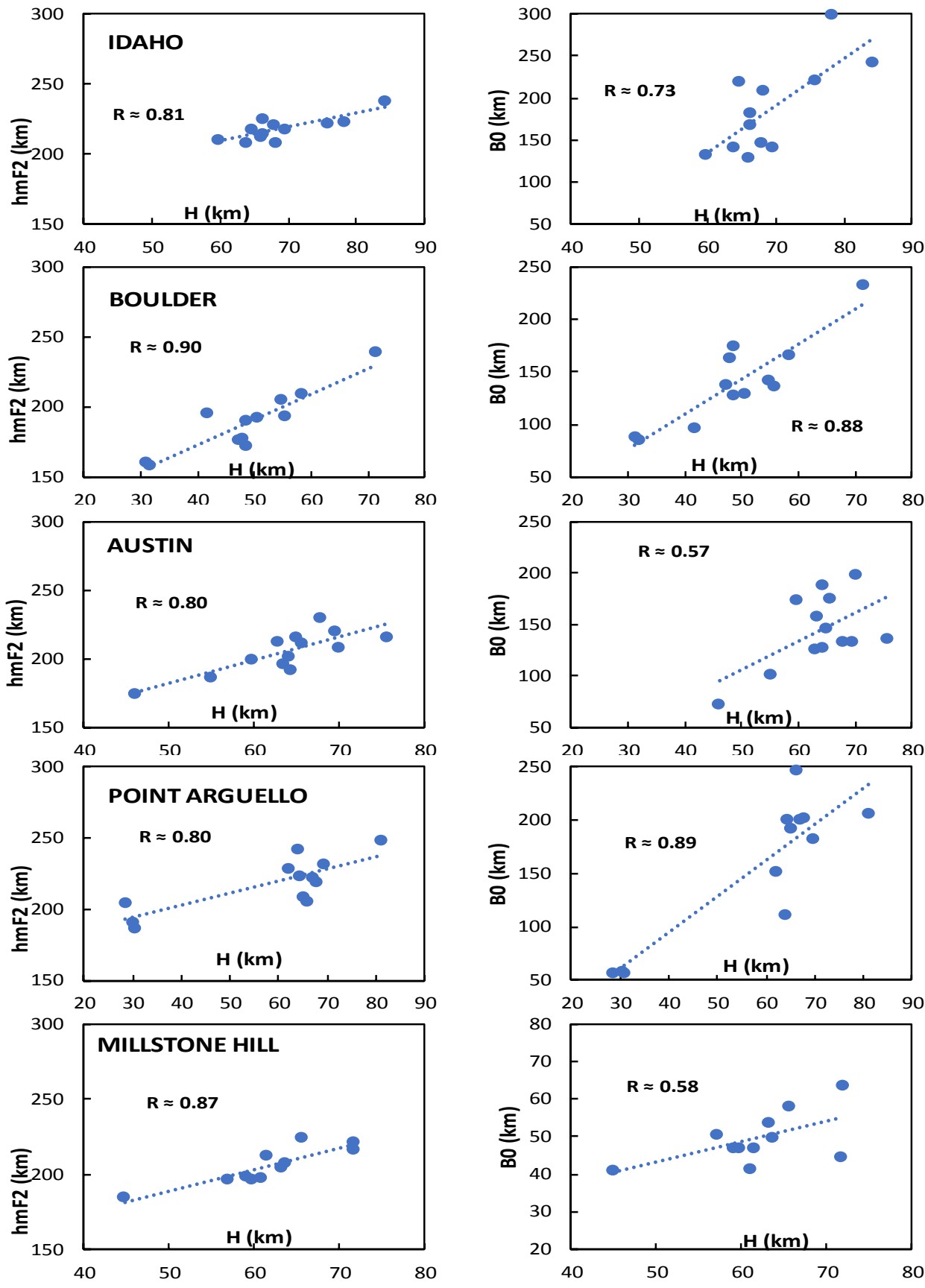

**Figure 5:** Linear relationship of H versus *hmF2* and H versus *B0* during the eclipse of 21 August 2017 progression phase.

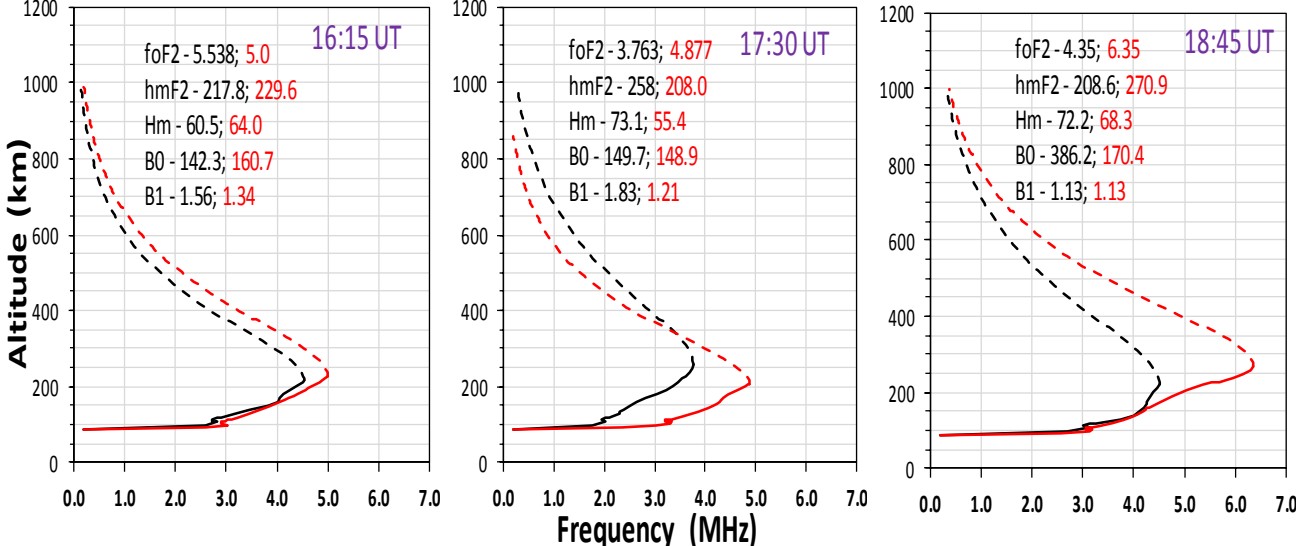

**Figure 6:** Example of the ionospheric profile at the eclipse window of Idaho National Lab showing the bottomside profile (continuous line) and the modelled topside profile shown as a dashed line. The maximum point of the continuous line is the point in which the peak value of the measured foF2 and hmF2 are obtained. The respective measured values foF2, hmF2 and the corresponding B1, B0, and Hm parameters values are provided in the plot. The black curve represents the profile for the eclipse day (August 21) and the red curve is for the one of the selected reference days, August 27. On the left side, was the profile during the first contact of the eclipse, the middle and the right profiles are for the maximum contact and the last contact of the eclipse respectively.