# Peer review of "SOLAR ECLIPSE-INDUCED PERTURBATIONS AT MID-LATITUDE DURING THE 21 AUGUST 2017 EVENT"

_Annales Geophysicae, 2018_

## Short Comment (SC1) · 19 May 2018

I'm interested in your paper, but I have some questions. can you help me? thank you very much!

1. I cannot understand the figure 3, it shows the deviation of NmF2 and hmF2 magnitudes duing the eclipse, why is it a contour rather than line?

2. Line 229-230: How could you assure the direction of the wind and its reversal?

---

## Short Comment (SC2) · 21 May 2018

Dear Dr. Q. LI, Thank you very much for your interest in our work. Since the paper is still at discussion stage, your comments on improving the standard of the manuscript are highly appreciated. Below are the responses to your questions.

SC1: I cannot understand the figure 3, it shows the deviation of NmF2 and hmF2 magnitudes during the eclipse, why is it a contour rather than line?

Response to SC1: Although observing the variation of NmF2 and hmF2 alone can be used, if properly analysed, for observing the changes in the behaviour of the thermospheric compositions and wind, but it is more convenient to describe these mechanisms by standardizing the original variable during the event. The normalization effort

(with the use of DNmF2 and DhmF2) presents the original variation of NmF2 and hmF2 onto directions which maximize the variance. Then, the result can be used for analyses of any mechanisms that drive the ionospheric plasma, if properly related.

The contour plot is to control the information about the mechanisms we are reporting. Using line plot is not bad either, but contour plot is preferable better than line plot.

SC2: Line 229-230: How could you assure the direction of the wind and its reversal?

Response to SC2: To identify the direction of the wind the DNmF2 colour legend was used, the negative values represent a westward wind contribution and the positive values is for the eastward wind. Looking at the marked eclipse region in the figure, one will see that the DNmF2 started decreasing from the first contact of the eclipse and maximized few minutes after the totality mark and started increasing again. It has been established that at daytime, the peak height of the plasma will be reduced due to lost in recombination, but at nighttime, equatorward neutral wind drives the F2-layer plasma to higher altitudes where ion loss rate is slower. The behaviour of the F2 plasma during solar eclipse cannot be completely related to the nighttime period due to the fact that all the processes controlling the nighttime variation are not completely actualised but can be related to partial nighttime/sunset period (see Adekoya et al., 2015). Thus, the slight increase in the peak height and equatorward neutral wind flow drives during the solar eclipse. The neutral wind acts jointly with the plasma flows from the topside ionosphere, resulting in F2 region plasma density variation. Therefore, the westward/eastward neutral wind flow was related to the depletion/enhancement in the deviation, which was clearly shown in the marked eclipse region of the figure. This makes the use of the deviation an important factor in analysing the thermospheric composition and neutral wind flow.

Reference: Adekoya, B. J., Chukwuma, V. U., and Reinisch, B. W.: Ionospheric vertical plasma drift and electron density response during total solar eclipses at equatorial/low latitude, J. Geophys. Res, 120, 8066-8084. doi:10.1002/2015JA021557, 2015

Thanks Adekoya, B. J. (for the authors)

---

## Referee Comment (RC1) · L. R. Cander (Referee) · 5 Jun 2018

SOLAR ECLIPSE-INDUCED PERTURBATIONS AT MID-LATITUDE DURING THE 21 AUGUST 2017 EVENT by B Adekoya

This manuscript attempts to provide a discussion related to the observed solar eclipse-induced perturbations at the mid-latitude during the 21 August 2017. Although long description of this event and conclusions reached are supported by data analysis, the obvious question is what is really new in author's results and findings which have not already been reviled in the large number of the reference papers.

Without any willingness to be negative in this report, it is necessary to draw attention on the following issues:

[Figure]

PP 39-42: In the context of the sentence "Different physical mechanisms (e.g. neutral wind, thermospheric composition, diffusion process etc.) that explain the distribution of plasma at the different ionospheric layers are well established", the subsequent sentence "However, these mechanisms do compete with themselves in explaining other layers, especially for the topmost F2 layers", is confusing. Particularly, who are "other layers" and where is "topmost" F2 layer?

PP46: "However", should be deleted, and star the sentence simple - At equatorial and low-latitudes...

PP73: foF2 is an ionospheric characteristic not an ionospheric parameter

PP272-273: The sentence "This paper presents the induced perturbation of solar eclipse of 21 August 2017 on the ionospheric F parameters and their behaviour in predicting one another at mid-latitude" is not clear to me because I could not understand who is predicting what and where are the results of that prediction. See also the first sentence in Abstract and PP282

PP276-277: There is not such name as "the F layer ionosphere". As authors know very well, there are F1 and F2 layers or F region of the Earth's ionosphere. See also PP281

Most importantly for the essence of the paper the last paragraph 252-267 is completely irrelevant. Furthermore, the IRI model is not generated to capture the conditions of the ionosphere during solar eclipse. See also PP285-287 as well as the last sentence in Abstract.

Although I am not a native English speaker, I feel free to suggest another careful proof-read to avoid some minor typo and language errors. For example:

PP80: NmF2 and hmF2- non italic; PP:84 NmF2, hmF2 – italic. See also a few more cases in the text;

PP:214: (NmF2e – NmF2c)/NmF2c x 100 should be 100 x (NmF2e – NmF2c)/NmF2c

Finally in my opinion this manuscript may become acceptable after major revisions and must be reviewed again.

---

## Author Comment (AC1) · 12 Jun 2018

Dear L. R. Cander, We thank you for the useful and supportive corrections. We believe that all suggestions made have been considered accordingly in this revised edition (not attached here) of the manuscript. Major corrections have been effected accordingly, and are highlighted (colour red) in the manuscript text. We have modified the manuscript accordingly, and the detailed corrections are listed below point by point:

Major concerns: Comment 1   This manuscript attempts to provide a discussion related to the observed solar eclipse induced perturbations at the mid-latitude during the 21 August 2017. Although long description of this event and conclusions reached are supported by data analysis, the obvious question is what is really new in author's

[Figure]

results and findings which have not already been reviled in the large number of the reference papers.

Response to Comment 1 ïČij The new findings of this present work have been given in full detail in the body text (line 294 - 300) and thus summarized in the abstract (see line 18 – 26). Moreover, the study of the circumstances of solar eclipse at the topside ionosphere and its plasma distribution mechanisms using the bottomside parameters, scale height and the F2-layer parameters makes it significantly different from previous studies (see line 52 – 56).

Comment 2   PP 39-42: In the context of the sentence "Different physical mechanisms (e.g. neutral wind, thermospheric composition, diffusion process etc.) that explain the distribution of plasma at the different ionospheric layers are well established", the subsequent sentence "However, these mechanisms do compete with themselves in explaining other layers, especially for the topmost F2 layers", is confusing. Particularly, who are "other layers" and where is "topmost" F2 layer?

Response to Comment 2 ïČij The statement has been rewritten (now in line 36-39)

Comment 3   PP46: "However", should be deleted, and star the sentence simple - At equatorial and low-latitudes...

Response to Comment 3 ïČij The "However" has been deleted and the correction has been made (see line 43)

Comment 4   PP73: foF2 is an ionospheric characteristic not an ionospheric parameter

Response to Comment 4 ïČij Thank you for this observation, the correction has been effected (see lines 72-73)

Comment 5   PP272-273: The sentence "This paper presents the induced perturbation of solar eclipse of 21 August 2017 on the ionospheric F parameters and their behaviour in predicting one another at mid-latitude" is not clear to me because I could

not understand who is predicting what and where are the results of that prediction. See also the first sentence in Abstract and PP28.

Response to comment 5 ïČij This sentence has been clarified in both in the text and the abstract (see line 13-14 and line 267)

Comment 6 • PP276-277: There is not such name as "the F layer ionosphere". As authors know very well, there are F1 and F2 layers or F region of the Earth's ionosphere. See also PP281.

Response to Comment 6 ïČij Thank you for this observation. This has been corrected in the text.

Comment 7 • Most importantly for the essence of the paper the last paragraph 252-267 is completely irrelevant. Furthermore, the IRI model is not generated to capture the conditions of the ionosphere during solar eclipse. See also PP285-287 as well as the last sentence in Abstract.

Response to comment 7 ïČij This aspect has been completely removed from the manuscript as suggested.

Comment 8 • Although I am not a native English speaker, I feel free to suggest another careful proofread to avoid some minor typo and language errors. For example: PP80: $NmF2$ and $hmF2$- non italic; PP:84 $NmF2$, $hmF2$ – italic. See also a few more cases in the text; PP:214: $(NmF2e – NmF2c)/NmF2c \times 100$ should be $100 \times (NmF2e – NmF2c)/NmF2c$

Response to Comment 8 ïČij Careful proofread of the manuscript has been carried out and all the language and typographical errors were corrected. For example: PP80 and PP84: $NmF2$ and $hmF2$ are now italicized (now in line 79 and 80). The text in PP214(Line 221) has been corrected accordingly.

Best regards, Adekoya, Bolarinwa J. (For the Authors)

---

## Referee Comment (RC2) · Anonymous Referee #2 · 19 Jun 2018

The manuscript is a good attempt to use NmF2, hmF2, B0, B1 and H to analyse ionospheric perturbation of solar eclipse. However, it is not well organized and written. In my opinion, the paper only can be reviewed again after major revisions. My specific suggestions are shown below.

Line 15, please give a brief introduction to "GIRO database", or at least give the full name of GIRO, otherwise it is difficult to know what kind of ionospheric parameters are used in your research.

Line 15, it is weird to use "percentage obscuration". In my opinion, the percentage of obscuration or the obscuration percentage is better. Similar to Line 211 and Line 213, there are "percentage concentration of the components" and "percentage deviation"

[Figure]

Line 22, "Need for IRI model to capture eclipse caused perturbation", it is not a complete sentence. Further, line 255-267, the authors said "IRI model doesn't capture the conditions of the ionosphere during solar eclipse", but didn't show any figure or table to support this judge. And I don't think IRI is a good tool to study ionospheric variations during solar eclipse.

Line 78-79, the authors said "The control day value is the mean of the values obtained on respective days ..." Specifically, which days did you choose as the control day? Was there geomagnetic storms in that period of time? Did you get the mean of the values by weighting?

Line 241-242, the authors said "The only exception ... at Millstone ... H versus B0 ..." however, it is clear that R is also low for the two figures of IDAHO.

Figures 1 and 2, for hmF2, scale height, bottomside, the variations of them are not very clear, especially at the stations of Eglin AFB, Boulder and Millstone Hill. I mean the noise is too large to get the valuable information. So it is a little far-fetched to draw your conclusion in "3 Result and Discussion".

Line 273-276, as the authors said, "ionospheric F2 parameters (NmF2 and hmF2), the bottomside profile thickness (B0) and shape (B1) parameters of electron density and the plasma scale height (H), which are not often used for eclipse study", so have you considered that why these parameters are seldom used in eclipse study? I guess that is because the useful information is probably covered by the noise, especially for such parameters as hmF2, B0, B1 and H.

Figure 3, how did you get this figure? I mean, for a certain electron density profile, there is only one NmF2 and hmF2. You know, NmF2 is F2 layer peak electron density and hmF2 is F2-layer peak density height. But in figure 3, it is very confusing that DNmF2 is varying with the change of DhmF2. I guess you mean Ne and corresponding height. Maybe my understanding is wrong, Please explain this further for helping readers understand this clearly.

In abstract and conclusion, the authors said "predicting one another". However, in the body of this manuscript, I didn't find which parameter is predicted. More importantly, the correlation between these parameters is not strong enough to predict each other. So it is not proper to judge that "Hence their relationship in predicting one another is established" If the authors want to prove that these parameters are predictable, they should provide some supporting figures or tables, instead of a very indiscreet sentence.

---

## Author Comment (AC2) · 29 Jun 2018

GENERAL RESPONSE We thank the reviewer for the useful and supportive corrections. We believe that all suggestions made by the reviewer have been considered accordingly in this revised edition of the manuscript. Major corrections have been effected accordingly, and are highlighted (color red) in the manuscript text. We have modified the manuscript accordingly, and the detailed corrections are listed below point by point:

Major concerns: Comment 1 • Line 15, please give a brief introduction to "GIRO database", or at least give the full name of GIRO, otherwise it is difficult to know what kind of ionospheric parameters are used in your research

Response to Comment 1 ïČij The GIRO means Global Ionospheric Radio Observatory database. This has been corrected in the manuscript (see page 1, line 15.

Comment 2 • Line 15, it is weird to use "percentage obscuration". In my opinion, the percentage of obscuration or the obscuration percentage is better. Similar to Line 211 and Line 213, there are "percentage concentration of the components" and "percentage deviation"

Response to Comment 2 ïČij The corrections have been effected now in lines 15 and 213

Comment 3 • Line 22, "Need for IRI model to capture eclipse caused perturbation", it is not a complete sentence. Further, line 255-267, the authors said "IRI model doesn't capture the conditions of the ionosphere during solar eclipse", but didn't show any figure or table to support this judge. And I don't think IRI is a good tool to study ionospheric variations during solar eclipse

Response to Comment 3 ïČij All IRI related statements in the manuscript have been deleted as suggested by Reviewer 1.

Comment 4 • Line 78-79, the authors said "The control day value is the mean of the values obtained on respective days . . ." Specifically, which days did you choose as the control day? Was there geomagnetic storms in that period of time? Did you get the mean of the values by weighting?

Response to Comment 4 ïČij The control day value is the average value of the two days before/after the eclipse day (i.e. 6, 12, 24 and 27). These reference days were chosen such that they have similar geomagnetic, interplanetary and solar properties with the eclipse day. The daily average value of the reference days and eclipse day for interplanetary index (Ap and ), and solar flux unit index (F10.7) ranges 8 – 12 nT for Ap, 20 – 27 nT for and 75.6 – 89.1 sfu (1 solar flux unit (sfu) = 10⁻$^{22}$ Wm⁻$^2$ Hz⁻Âż) for F10.7, indicating that geomagnetic and solar activities of these days is

unsettled (see Adekoya et al., 2015 for classification of the activities). This is because under the same classifications, the effect of eclipse in the ionosphere is expected to be noticed when compared with the control day. The calculated daily average of summation Kp, Ap and solar flux indices was obtained from the National Space Science data Centres (NSSDC's) OMNI database https://omniweb.gsfc.nasa.gov/. This point has been included in the manuscript (see line 79 - 87)

Comment 5 • Line 241-242, the authors said "The only exception . . . at Millstone. . . H versus B0 . . ." however, it is clear that R is also low for the two figures of IDAHO.

Response to comment 5 ïČij Thank you for the observation. The statement has been corrected accordingly (see line 262-264).

Comment 6 • Figures 1 and 2, for hmF2, scale height, bottomside, the variations of them are not very clear, especially at the stations of Eglin AFB, Boulder and Millstone Hill. I mean the noise is too large to get the valuable information. So it is a little far-fetched to draw your conclusion in "3 Result and Discussion".

Response to Comment 6 ïČij After critical observation of the said figures panels, the authors observed that there are no noise in the variations of the parameters during the eclipse window, rather the effect of eclipse was noted in comparison with the control day. Moreover, the digital ionosonde data used were from GIRO, which have minimal/negligible level of noise in the data records (see Reinisch and Galkin 2011; Reinisch et al., 2018). In addition, the reference days were chosen (as explained in Response to Comment 4) in a way that the data are not contaminated by noise, if there is any.

Comment 7 • Line 273-276, as the authors said, "ionospheric F2 parameters (NmF2 and hmF2), the bottomside profile thickness (B0) and shape (B1) parameters of electron density and the plasma scale height (H), which are not often used for eclipse study", so have you considered that why these parameters are seldom used in eclipse study? I guess that is because the useful information is probably covered by the noise,

especially for such parameters as hmF2, B0, B1 and H

Response to comment 7 ïČij The use of the parameters in this study is a novel way of observing the ionospheric behavior at lower and topside ionosphere during solar eclipse, and as explained in Response to Commment 6, it is not associated with noise.

Comment 8 • Figure 3, how did you get this figure? I mean, for a certain electron density profile, there is only one NmF2 and hmF2. You know, NmF2 is F2 layer peak electron density and hmF2 is F2-layer peak density height. But in figure 3, it is very confusing that DNmF2 is varying with the change of DhmF2. I guess you mean Ne and corresponding height. Maybe my understanding is wrong, Please explain this further for helping readers understand this clearly.

Response to Comment 8 ïČij The clarification of Figure 3 has been explained to aid the readers' curiosity and understanding in line 221-255 and under figure caption in line 445 – 450.

Comment 9 • In abstract and conclusion, the authors said "predicting one another". However, in the body of this manuscript, I didn't find which parameter is predicted. More importantly, the correlation between these parameters is not strong enough to predict each other. So it is not proper to judge that "Hence their relationship in predicting one another is established" If the authors want to prove that these parameters are predictable, they should provide some supporting figures or tables, instead of a very indiscreet sentence.

Response to comment 9 ïČij We agree with your submission, and in line with the other reviewer's suggestion, we have deleted appropriately and the sentences have been rewritten both in the abstract and conclusion (see line 13-14 and line 275)

---

## Referee Report (RR1)

Referee Report for
**Solar Eclipse-Induced Perturbations at Mid-Latitude During the 21 August 2017 Event**
Submitted to Annales Geophysicae by Adekoya et al., 2018 (Manuscript #angeo-2018-35)

*Adekoya et al., 2018* presents observations of ionospheric impacts of the 21 August 2017 Total Solar Eclipse using measurements from a network of ionosondes across the United States. The authors then relate these observations back to theory by fitting the observations to a Chapman-type ionosphere. This paper has the potential for being a good contribution to the literature by relating observations during a highly-publicized eclipse event to a well known ionospheric model. However, I believe that the manuscript requires substantial revision before it can be published in *Annales Geophysicae.*

**Major Comments**

1. Section 2 (Data source, methodology, and path of the eclipse) would benefit greatly by being expanded and broken into sections. Suggestions include:
   a. Add a figure that shows the path of the eclipse, the percertage of maximum obscuration, and location of the ionosondes. This is especially important as you cannot guarantee that the websites you list for path information will always exist.
   b. Add a figure using actual data from the event showing how you fit the Chapman profile, and identify the parameters derived ($H, B_0, B_1$). Add text explaining how this allows you to draw conclusions relating the topside ionosphere to bottomside measurements.
2. Figures 1 & 2:
   a. I don't understand why you ordered the ionosondes in the manner that you did. Could you please order them from west (top) to east (bottom)? This is also the order in which the eclipse progressed across the UT.
   b. Instead of using LT as your X-axis, try using time relative to eclipse maximum, with 0 in the center. This way, it will be easy to compare the effect at all stations. To show local time, add another dashed or dotted line to each panel showing the local solar zenith angle. Put the solar zenith angle on the right-hand y-axis.
   c. In the caption, add some text guiding the reader of what eclipse signature to look for and why.
3. Figure 3
   a. The biggest problem here is that the Point Argello panel is dominantly green, but there are no green values in the colorbar. This absolutely must be fixed. For the colorbar, consider using a symmetric, diverging colormap. Say, red-white-blue (like below) with the range -65% to +65%.

[Figure]

bwr

b. I'd recommend fixing the Y-axis to some symmetrical value (say +/- 25) for all stations for easy comparison.
c. As for figures 1 & 2, I'd recommend plotting the x-axis in hours relative to eclipse maximum.
d. Order the panels from west to east (or some other way that makes sense according to what you are trying to show).

4. There are numerous grammatical errors. Some are minor, but some are major. For example, line 284 does not make sense. It currently reads, "Hence their relationship in describe one another is established."

**Minor Comments**

1. Line 56: "This," → "Thus,"
2. Line 59: "result and discussion were" → "results and discussion are"
3. Line 83: Kp is unitless (not nT)
4. Line 152: "recombination too" → "recombination, too"
5. Line 200: "This imply" → "This implies"

*Thank you for your submission. Good luck with the revisions!*

---

## Author Response (AR3)

**RESPONSE TO ANONYMOUS REVIEWER 3 COMMENTS**

MS No.: angeo-2018-35
MS Type: Regular paper
Iteration: Minor Revision
MANUSCRIPT TITLE: **Solar Eclipse-Induced perturbations at mid-latitude during the 21 August 2017 event**

**Dear Editor,**

**GENERAL RESPONSE**
We thank the reviewer for the useful and supportive corrections/suggestions. We believed that all suggestions made have been attended to accordingly. Major corrections have been carried out and are highlighted (red coluor) in the manuscript. More figures have been added as suggested, this has made the initial figure numbers to be reordered.  The table has been edited to highlight the corresponding local time to the universal time  of the solar eclipse progression. all additional references quoted in the text had been included in the reference list and highlighted in red colour.

Further, the names of the authors appeares only because of their contributions in planing, data analysis and writing the report. Therefore, two more authors who participated largely in the review processes from the first submission to present have bneen added. The Authors list now reads " *Bolarinwa J. Adekoya, Babatunde O. Adebesin, Victor. U. Chukwuma, Timothy W. David, Stephen O. Ikubanni, Shola J. Adebiyi, and Olawale. S. Bolaji*" based on their contributions. The third author was/is my Ph.D supervisor/mentor who have contributed majorly to my academic growth till present. The last author is a senior colleague that rendered his help when we had problem in some key area during the revisions. Not only that, he partook in the general proofreading/editing of the manuscript.

The authors are grateful for the useful comments and suggestions on the structure of the manuscript. This has tremendously improved the standard of the manuscript. We have modified the manuscript accordingly, and the detailed corrections are listed below point by point.

**GENERAL COMMENT:**
**Adekoya et al., 2018 presents observations of ionospheric impacts of the 21 August 2017 Total Solar Eclipse using measurements from a network of ionosondes across the United States. The authors then relate these observations back to theory by fitting the observations to a Chapman type ionosphere. This paper has the potential for being a good contribution to the literature by relating observations during a highly-publicized eclipse event to a well known ionospheric model. However, I believe that the manuscript requires substantial revision before it can be published in Annales Geophysicae.**

**Major comments:**
*Comment 1*
- **Section 2 (Data source, methodology, and path of the eclipse) would benefit greatly by being expanded and broken into sections. Suggestions include:**
a. **Add a figure that shows the path of the eclipse, the percentage of maximum obscuration, and location of the ionosondes. This is especially important as you cannot guarantee that the websites you list for path information will always exist.**
   **Response to Comment a**
   - ✓ We have added an orthographic projection map showing the coverage area and the circumstances of the solar eclipse, see figure 1 and line 65 - 66.
b. **Add a figure using actual data from the event showing how you fit the Chapman profile, and identify the parameters derived (H, B0, B1). Add text explaining how this allows you to draw conclusions relating the topside ionosphere to bottomside measurements.**
   **Response to Comment b**
   - ✓ We have added a profiler (Figure 6) that illustrates the relationship between the bottomside and the topside parameters, using the actual data of the event and the corresponding non-eclipse period, and texts explaining how the Chapman profile can be fiited (see line 92 – 104 and line 281 – 289).

*Comment 2*
- **Figures 1 & 2:**
a. **I don't understand why you ordered the ionosondes in the manner that you did. Could you please order them from west (top) to east (bottom)? This is also the order in which the eclipse progressed across the UT.**

   **Response to Comment a**
   - ✓ The ionosonde stations have been reordered according to the eclipse progression time.

b. **Instead of using LT as your X-axis, try using time relative to eclipse maximum, with 0 in the center. This way, it will be easy to compare the effect at all stations. To show local time, add another dashed or dotted line to each panel showing the local solar zenith angle. Put the solar zenith angle on the right-hand y-axis.**

   **Response to Comment b**
   - ✓ We did used the local time (LT) corresponding to the start (S), maximum contact (M) and last contact (E) of the eclipse, which made it more easy to compare the effect locally. In Table 1, both the universal time (UT) and the local time progression of the event have been provided. The UT and LT difference for each station is also presented. All these will help the reader to easily relates the eclipse effect at easch station.

c. **In the caption, add some text guiding the reader of what eclipse signature to look for and why.**

   **Response to Comment c**
   - ✓ Text on the expected signature has been added.

*Comment 3*
- **Figure 3**
a. **The biggest problem here is that the Point Argello panel is dominantly green, but there are no green values in the colorbar. This absolutely must be fixed. For the colorbar, consider using a symmetric, diverging color map. Say, red-white-blue (like below) with the range -65% to +65%.**

[Figure]

   **Response to Comment a**
   - ✓ We really appreciate the reviewer for this wonderful correction. We prudently carried out the correction as suggested and the end result was astonished and well represented (see Figure 4).

b. **I'd recommend fixing the Y-axis to some symmetrical value (say +/- 25) for all stations for easy comparison.**

   **Response to Comment b**
   - ✓ The Y-axis has been fitted, using symmetry interval unit value, because maximum value along Y-axis is not symmetry for all the plots. And if we try to set it above the maximum, it will only show open spaces that will raise some doubting questions in the mind of the readers.

c. **As for figures 1 & 2, I'd recommend plotting the x-axis in hours relative to eclipse maximum.**

   **Response to Comment c**
   - ✓ The figures were plotted for the daytime periods (i.e., from sunrise to sunset period). Having understand the contribution of photoionization to the plasma distribution in the ionosphere, solar eclipse present the consequences of the sudden lost in the photoionization during the daytime. Therefore, it will be advisable to observe the state of the ionosphere before and after the eclipse for better understanding of its problems and morphology. Moreso, it will inform the readers the clear picture of the effect of reduction in photoionization.

d. **Order the panels from west to east (or some other way that makes sense according to what you are trying to show).**

**Response to Comment c**
- ✓ This, we had attended to (see comment 2a)

*Comment 4*
- **There are numerous grammatical errors. Some are minor, but some are major. For example, line 284 does not make sense. It currently reads, "Hence their relationship in describe one another is established."**
  **Response to Comment**
  - ✓ Careful proofread of the manuscript has been done and all the language and typographical errors were corrected.
  - ✓ We have checked and modified the statement accordingly (see line 307 – 308).

- **Minor Comments**
- **1. Line 56: "This," à "Thus,"**
  - ✓ Corrected (see line 55)
- **2. Line 59: "result and discussion were" à "results and discussion are"**
  - ✓ Corrected, now reads results and discussion (line 58).
- **3. Line 83: Kp is unitless (not nT)**
  - ✓ This was a mistake, now corrected (see line 83).
- **4. Line 152: "recombination too" à "recombination, too"**
  - ✓ Corrected, now in line 163
- **5. Line 200: "This imply" à "This implies"**
  - ✓ Corrected, now in line 215

Best regards,

ADEKOYA, B. J.(Ph.D)
For the Authors

**RESPONSE TO ANONYMOUS REVIEWER 2 COMMENTS**

**MS No.: angeo-2018-35**
**MS Type: Regular paper**
**Iteration: Minor Revision**
MANUSCRIPT TITLE: **Solar Eclipse-Induced perturbations at mid-latitude during the 21 August 2017 event**

_Dear Editor,_

_GENERAL RESPONSE_
We thank the reviewer for the useful and supportive corrections/suggestions. We believed that all suggestions made by the reviewers' have been attended to accordingly. Major corrections have been carried out and are highlighted (red coluor) in the manuscript. More figures have been added as suggested by another reviewer, this has made the initial figure numbers to be reordered. The table has been edited to highlight the corresponding local time to the universal time  of the solar eclipse progression. all additional references quoted in the text had been included in the reference list and highlighted in red colour.

Further, the names of the authors appeares only because of their contributions in planing, data analysis and writing the report. Therefore, two more authors who participated largely in the review processes from the first submission to present have bneen added. The Authors list now reads " _Bolarinwa J. Adekoya, Babatunde O. Adebesin, Victor. U. Chukwuma, Timothy W. David, Stephen O. Ikubanni, Shola J. Adebiyi, and Olawale. S. Bolaji_" based on their contributions. The third author was/is my Ph.D supervisor/mentor who have contributed majorly to my academic growth till present. The last author is a senior colleague that rendered his help when we had problem in some key area during the revisions. Not only that, he partook in the general proofreading/editing of the manuscript.

The authors are grateful for the useful comments and suggestions on the structure of the manuscript. This has tremendously improved the standard of the manuscript. We have modified the manuscript accordingly, and the detailed corrections are listed below point by point:

**Major concerns:**
_Comment 1_
- **For Comment 2 of first version of manuscript, the authors corrected "percentage obscuration" but didn't correct "percentage deviation" in the manuscript.**
   **Response to Comment 1**
      ✓   This is an oversight and have now corrected in the edited version

_Comment 2_
- **In Figures 1 and 2, it is clear that NmF2 goes down during eclipse time, but for hmF2, scale height, bottomeside, the curves fluctuate sharply, with a lot of spikes, so it is hard to draw a convincing conclusion.**
   **Response to Comment 2**
      ✓   We have replotted the figures to clearly  show the morphology of the ionospheric characteristics durng the eclipse window.

_Comment 3_
- **Line 141-143, the authors said "The ionosphere over Eglin AFB, Boulder, Point Arguello, Millstone Hill and Idaho National Lab, did not show any contrary variation to that observed at Austin during the eclipse event. The decrease and increase in NmF2 and hmF2 after the maximum magnitude was simultaneous."**
   **But I don't agree with it. First, I didn't find that NmF2 and hmF2 change simultaneouly. Second, I don't think the variations between these stations are consistent. Please analyse these stations one by one. For example, for NmF2, it is really clear that it goes down from S to M and goes up from M to E. However, for hmF2, it is totally different, as shown in the table below.**

   **The variation of hmF2 during eclipse compared to control day**
   **Station from S to M from M to E**

**Austin Up Down**
**Eglin AFB Not available Not available**
**Boulder No apparent trend Down**
**Point Arguello Up A valley**
**Millstone Hill No apparent trend Down**
**Idaho National Lab No apparent trend DownPP46: "However", should be deleted, and star the sentence simple - At equatorial and low-latitudes...**

**Response to Comment 3**
- ✓ We appreciate the reviewer for this observation, this has tremendously improved our data analysis. The discrepancies have now been corrected and the figure have been replotted. The figures are more clearler and explained the simultaneity in the ionospheric variations observed at the eclipse window better.

*Comment 4*
- **Line 175-176, the authors said "It was observed from the plots that the minimum decrease in NmF2 amplitude corresponds to increase in H at all stations"**
  **But at least at Austin and Eglin AFB, I didn't find the conclusion above.**

- **Line 205-206, the authors said "B1 responded with a decrease at the first contact of the eclipse compared to the control day."**
  **But at least at Austin, Eglin AFB, Milestone Hill, Idaho National Lab, I didn't find the conclusion above.**

- **Line 207-209, the authors said "B0 parameter from the first contact increases and reached the maximum peak few minutes after the maximum obscuration magnitude, which coincided with the minimum decrease in B0."**
  **But at least at Austin, Eglin AFB, Milestone Hill, I didn't find the conclusion above.**

**Response to Comment 4**
- ✓ As said earlier, all the figures have been regenerating. Kindly check the highlited texts in the reviewed manuscript

*Comment 5*
- **In Figure 1, there are no available data during and after eclipse at Eglin AFB. However, in Figure 3, the percentage of deviation at Eglin AFB is complete. How did you get this?**

**Response to comment 5**
- ✓ The data gap in Eglin AFB, Figure 4 (old figure 3) was interpolated by the software used in generating the plot. However, the Eglin AFB plot has been removed from Figure 4 for the benefit of doubt on the description of the ionospheric behavior over the station.

*Comment 6*
- **I have to say, I haven't yet understood Figure 3. I guess in this figure, DNmF2 is the ratio of the electron density at different height between eclipse day and control day, that is to say, it may be ((Ne)e - (Ne)c)/(Ne)c) x 100, but the authors said "was defined as the ratio of ((NmF2e - NmF2c)/NmF2c) x 100." So I am really confused by DNmF2. For a certain local time at one station, there are only one NmF2e and one NmF2c.**

**Response to Comment 6**
- ✓ We belived that figure 4 (old figure 3) was used to described the neutral wind flow and thermospheric compositions during solar eclipse (see line 235 – 273). The $\delta NmF2$ is the percentage deviation of the maximum electron density at the F2 layer, while the Ne is the electron density at different altitude of the ionosphere. Whereas, NmF2e and NmF2c are the maximum electron desity of the F2 layer during the eclipse and non-eclipse (i.e., control period, which is the average of the respected non-eclipse days). Indeed, for a certain period of local time there is only one NmF2e and NmF2c, for every period of the day, the the NmF2 plot is plotted. However, we have plotted in figure 6, the electron density profile that explain the distribution of the electron at different altitude during solar eclipse and the corresponding non-eclipse period.

*Comment 7*

- **Moreover, in Figure 3, at Point Arguello, the colour scale didn't show the green.**

**Response to comment 7**
- ✓ In Figure 3 (now Figure 4) we have replotted the figure, changing the colour legend and the axis scaling to clearly shows the simultaneity in the colour distribution.

*Comment 8*
- **For Comment 5 of first version of manuscript, the reviewer asked a questions "Line 241-242, the authors said "The only exception … at Millstone… H versus B0 …" however, it is clear that R is also low for the two figures of IDAHO."**
  **But this time, in second version of manuscript, Line 262 to 264, the authors said "The only exception where low correlation was observed was at Idaho (0.47) and Millstone (0.37) with respect to the H versus B0 relationship."**
  **So the correlation at Idaho (0.52) with respect to H versus hmF2 is not an exception? In fact, in Figure 4, only at Point Arguello and Millstone Hill, the correlation is good enough.**

- **If the authors are not able to explain height parameters properly, just showing them directly without convincing processing and analysis, I don't think this manuscript is worth being published.**

**Response to Comment 8**
- ✓ All the discripancies have been corrected. Figure 4 (now Figure 5) has been replotted. More explanations have been provided in the text on the height parameters, which allows to drawn conclusions relating the topside ionosphere to bottomside measurements.

Best regards,

ADEKOYA, B. J.
For the Authors

[revised manuscript text omitted]